# A Modified Approach for Minimally Invasive Tubular Microdiscectomy for Far Lateral Disc Herniations: Docking at the Caudal Level Transverse Process

**DOI:** 10.3390/medicina58050640

**Published:** 2022-05-05

**Authors:** Murray Echt, Adewale Bakare, Richard G. Fessler

**Affiliations:** Department of Neurological Surgery, Rush University Medical Center, Chicago, IL 60612, USA; murray_a_echt@rush.edu (M.E.); adewale_a_bakare@rush.edu (A.B.)

**Keywords:** far lateral lumbar disc, minimally invasive spine surgery, tubular discectomy

## Abstract

*Background and Objectives*: The use of minimally invasive retractor systems has significantly decreased the amount of tissue dissection and blood loss, and the duration of post-operative recovery after far-lateral disc herniations (FLDH). In this technical note, the technique of docking the tubular retractor on the caudal transverse process is described for an efficient approach with a decreased need for manipulation of the exiting nerve root. *Materials and Methods*: The case reported is that of a woman affected by a right-sided FLDH at the L4–5 level causing an L4 radiculopathy with weakness and numbness. A review of the literature for FLDH regarding the key anatomy used during a far lateral approach was also performed. *Results*: The patient showed a significant improvement of her dorsiflexion weakness and radiating leg pain at her 2-week and 5-week post-operative visits, and at a 6-month follow-up she had near-complete relief of her symptoms, including resolution of foot numbness. Prior techniques for tubular microdiscectomy for FLDH report docking on the facet joint, pars interarticularis, and the cranial transverse process. *Conclusions*: This technical note details that the utility of docking a tubular retractor at the caudal transverse process improves upon already established techniques for minimally invasive tubular discectomy for FLDH.

## 1. Introduction

The use of minimally invasive retractor systems has significantly decreased the amount of tissue dissection and blood loss, and the duration of post-operative recovery for far-lateral disc herniations (FLDH) [1,2,3]. There are various techniques for docking the tubular retractor, including on the facet joint, pars interarticularis, and the cranial transverse process (TP) [4]. The challenges for a minimally invasive surgery for FLDH consist in limited bony landmarks compared to soft-tissue elements, which results in a variation of where to dock the retractor [5,6]. The anatomy is also less familiar than midline posterior approaches, potentially creating spatial confusion. Further, the dorsal root ganglion (DRG) of the exiting nerve root is more susceptible to thermal and mechanical injury, requiring less manipulation than what is used during standard discectomy [1,5].

In this technical note, the precise technique for docking the tubular retractor on the caudal TP is described for an efficient approach with a decreased need for manipulation of the exiting nerve root.

## 2. Materials and Methods

We have refined a new docking and exposure technique utilizing a minimally invasive tubular retractor to approach far lateral disc herniations more efficiently by docking at the transverse process at the caudal level. This is offered as a fine-tuning enhancement of a prior technique described by the senior author, where docking was performed at the cranial level and then angled caudal [7]. This tweak allows for an easy definition of key anatomy and a logical approach towards identifying the disc space and limiting the manipulation of the sensitive exiting nerve root and DRG. The case reported in our study is that of a woman affected by a right-sided far lateral disc herniation at the L4–5 level, who underwent the present procedure at Rush University Medical Center, Chicago, Illinois. The surgical procedure was performed by the senior author (R.G.F.). The patient read and signed the written informed consent form. Ethics approval was deemed unnecessary according to the hospital’s published Institutional Review Board. Additionally, this novel tweak is covered within the confines of the existing previously described approach to far lateral disc herniation without deviation from the standard of care [4].

### 2.1. Clinical Evaluation

The patient presented with a severe, 10/10 numeric rating scale of pain along the right buttock radiating to the anterior and medial aspect of her thigh into the anterior calf to the top of her foot. This was associated with numbness and weakness of her right foot dorsiflexion. The diagnosis was confirmed on the magnetic resonance imaging (MRI) sagittal T2 weighted sequence (Figure 1a) and axial T2 weighted sequence (Figure 1b), which demonstrates a far lateral right L4–5 disc herniation with cephalad migration. The patient underwent several weeks of non-operative treatments including a Medrol dosepak, nonsteroidal anti-inflammatories, and muscle relaxers with minimal temporary improvement in pain. Physical examination demonstrated a profound weakness of her right foot, including 3/5 strength of the tibialis anterior, an inability to heel walk, and a numbness along an L4 dermatome. Given the presence of a severe motor deficit in the subacute period and a strong desire to regain her foot strength, a joint decision made with the patient led to pursuing early surgery rather than engaging in additional non-operative treatment with epidural steroid injection and physical therapy [5]. The patient’s body mass index (BMI) was 25.5 kg/m^2^.

### 2.2. Technical Description

Following induction with general anesthesia and endotracheal intubation, the patient was turned onto the prone position on a Wilson Frame to maximize lumbar flexion. Fluoroscopy was placed in the lateral position, and the patient’s lumbar spine was prepped and draped in the usual sterile fashion. Using lateral fluoroscopy, an appropriate incision to approach the right L 4–5 level was identified and marked 6 cm lateral to midline. After local anesthetic infiltration and incision, the first dilator was placed through the incision and advanced to the L5 TP (Figure 2a). This position was confirmed fluoroscopically. Care must be taken not to use excessive downward force when docking on the TP to avoid incurring a fracture. ‘Walking’ the initial dilator medially and finding the border of the TP-facet junction, similar to a starting point for pedicle screw placement, allows for more downward force and is a useful guide for medial-lateral angulation. A series of dilators were then placed over this, followed by an 18 mm tubular retractor, which was positioned, angled medially, and locked in place (Figure 2b). This position was further confirmed fluoroscopically. Bovie electrocautery was used to remove a small amount of residual tissue at the bottom of the working channel (Figure 3a). An angled curette was then used to define the plane between the soft tissue and the superior foramen and ventral to the TP. Kerrison punch was used to perform a dorsal foraminotomy and to remove the caudal portion of the TP. The exiting nerve root was readily identified and gently retracted superiorly. The bulging disc was identified and confirmed fluoroscopically (Figure 3b). A 15-blade scalpel was used to incise the annulus, and a discectomy was performed using a series of curettes and pituitary rongeurs. The foraminal and epidural space was then explored for additional fragments using a series of blunt probes. At the completion of decompression, the exiting nerve root was noted to be well decompressed throughout its course.

### 2.3. Comprehensive Literature Review

A comprehensive review of the literature for FLDH regarding the key anatomy used during a minimally invasive far lateral approach was also performed. Articles on Pubmed were searched, and publications that contained a description of surgical techniques for FLDH were collected. Open mid-line approaches were not included. Articles that were collected were divided into a mini-open paramedian/Wiltse approach and tubular minimally invasive approaches. These were further sub-divided by the authors’ description of their key anatomic landmark for accessing the disc space and herniated fragment.

## 3. Results

### 3.1. Clinical Outcome

The patient showed a significant improvement of her dorsiflexion weakness and radiating leg pain at her 2- and 5-week post-operative visits, with the pain score reduced to 1/10. At the 6-month follow-up, she had near-complete relief from her symptoms, including resolution of foot numbness. Intermittently, she did continue to have right foot slapping when walking. Given the substantial improvement, no post-operative MRI was deemed necessary. The patient requested and was cleared to return to downhill skiing following this visit.

### 3.2. Comprehensive Literature Review Results

Articles that were collected included key anatomy/docking on the facet joint, pars interarticularis, and the cranial or caudal TP (Table 1).

## 4. Discussion

FLDH originate within or lateral to the neural foramen compressing the exiting nerve root. These patients are more likely to present with neurological deficits given the constrained space. Traditionally, these herniations were accessed through a midline approach with laminotomy, with partial resection of the pars or total facetectomy with fusion. Modified far-lateral approaches were successfully adopted to access FLDH with varied nuances of approach. As techniques advanced, minimally invasive approaches to FLDH have been advocated, including the use of tubular retractors. Subtle nuances to approaching FLDH have the potential to enhance the safety and efficiency of the procedure. Here, we present a modification to previously published work to improve upon already established techniques for minimally invasive tubular discectomy for FLDH.

The benefits of the minimally invasive approach to docking at the caudal TP described above emphasizes the minimal soft tissue dissection and bone removal necessary for FLDH. Previous reports include a description of the partial removal of the lateral pars. The lateral pars actually has the largest thickness, and its removal has a potential for iatrogenic stress fractures [18,19]. Similarly, surgeons that target the facet joint risk worse outcomes in cases of excess removal of even 25% of the joint [20]. Docking on the inferior TP also provides a readily identifiable anatomy on both x-rays and direct visualization, which decreases the need for more complex and expensive imaging such as intraoperative O-arm computerized tomography navigation [16,21]. If navigation-assistance is utilized, a virtual tip offset of variable length may be used to identify the entry point and surgical trajectory to dock along the caudal TP. Further, it eliminates an excess step compared to docking at the cranial TP, which requires subsequent caudal angulation, as was previously described by the senior author [7]. A decreased movement of the tubular retractor also reduces a possible error in over- or under-angulation, more readily isolating the disc space and herniated fragment. Contralateral approaches that utilize an inside-out technique require the limited boney removal of the base of the lamina and spinous process, however placing the operator at a disadvantage for extraforaminal disc herniations [22]. Extraforaminal lesions, or extreme lateral, should be reached by a far lateral approach described above.

Decreased nerve root manipulation is achieved via the natural corridor to the disc space by hugging the pedicle of the caudal level. The caudal TP naturally leads the surgeon medially to the junction with the facet. After release of the intertransverse membrane from the caudal TP, this TP-facet junction can then be used to slide against the bone cranially as it leads along the pedicle. This isolates the caudal aspect of the neural foramen and directly towards the disc space without need to strip soft tissue over the exiting nerve root [23]. Additionally, in a cadaveric study, it was found that drilling of the caudal TP, or in the case of L5-S1 FLDH the superior sacral ala, most easily exposed the exiting nerve root while also providing access to the disc without need for further boney removal [24]. In almost all cases, the extraforaminal disc fragment will be caudal to the exiting nerve root and displace it superiorly and laterally [5]. Thus, the caudal TP will be the most consistent boney landmark for accessing the extraforaminal disc. Extruded disc fragments may be freed with blunt dissecting probes, sliding beneath the nerve and the disc space. If necessary, the annulus may be incised for further disc removal. A minimal retraction of the exiting nerve root is required but may be performed in a lateral-cranial direction. This approach also puts this surgeon at proximity to the level of the disc, requiring less retraction of soft tissue. Other approaches place the disc at a further reach, including docking at the pars or facet.

Endoscopic approaches are well adapted for FLDH; however, they require a learning curve that may not be suitable for most spine surgeons. Tubular retractors are more commonly used [16], and may be easily substituted for a traditional mini-open Wiltse approach. Endoscopic technique utilizes a similar approach to our current description by docking at the junction of the caudal TP and facet, thus taking advantage of a decreased need for soft-tissue dissection and bony removal. Despite descriptions of similar approaches for FLDH in the literature, we believe our technical note adds to previously published work by modifying the same approach to a tubular minimally invasive discectomy. The senior author also wished to update his prior technical description [7].

Limitations include only a single patient presentation rather than a case series or potentially comparing results with previous technique when docking at the cranial level. However, the rarity of the FLDH inherently limits the number of patients, and therefore finding statistical significance between techniques via an under-powered study is unlikely. Recurrence of an FLDH poses an additional challenge not addressed herein, and optimal treatment is controversial even in conventional revision discectomy vs. fusion [25]. The advantages of docking at the caudal level TP are salient and without any other modifications to prior reports. This should allow for easy adoption and enhance patient outcomes. Future studies with large prospective case series with granular details of operative technique and anatomical approach are needed.

## 5. Conclusions

This technical note describes the advantages of docking at the caudal level TP during a tubular MIS approach for FLDH. Previously, techniques included docking on the facet joint, pars interarticularis, and the cranial TP. Utilizing this reported modification to prior reports requires no further modifications and should allow for an easy adoption. The utility of this approach improves upon already established techniques for MIS tubular discectomy for FLDH by minimizing soft tissue dissection, nerve root manipulation, and bone removal, easily identified anatomy on both x-ray and direct visualization. It also eliminates an excess step compared to docking at the cranial TP, which requires subsequent caudal angulation.

## Figures and Tables

**Figure 1 medicina-58-00640-f001:**
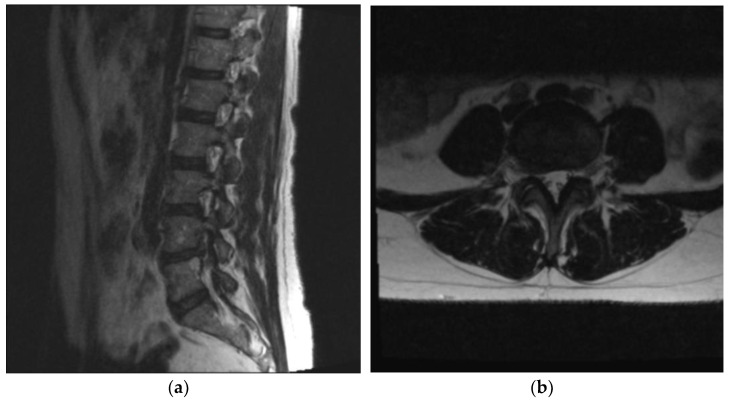
T2 weighted Magnetic Resonance Imaging (MRI). (**a**) Sagittal sequence demonstrating severe foraminal stenosis at L4–5 with cephalad disc extrusion; (**b**) axial cut at the L4–5 disc level also demonstrating right-sided far lateral disc herniation.

**Figure 2 medicina-58-00640-f002:**
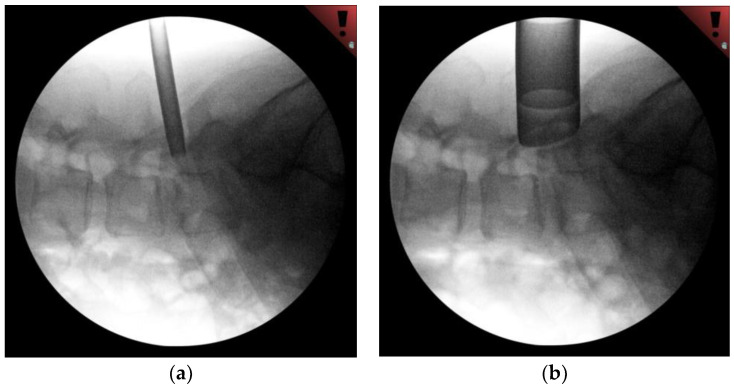
(**a**) Fluoroscopy of initial docking on the caudal TP with the initial dilator. (**b**) Fluoroscopy of the final position of the tubular retractor in line with the disc space, and the caudal TP remaining as the key bony element for docking.

**Figure 3 medicina-58-00640-f003:**
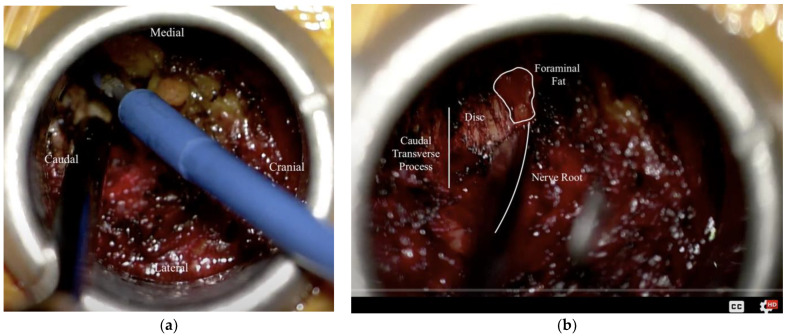
Intra-operative photo from a right-sided (ipsilateral) extra-foraminal approach to the L4–5 FLDH. (**a**) Exposure of the caudal transverse process and facet junction using the Bovie electrocautery; (**b**) Following the partial removal of the overlying intertransverse membrane attached to the caudal transverse process, immediately exposing the disc space, allowing for further exploration of the foramen (fat contained within the foramen is highlighted) as well as the exiting nerve root without need for complete stripping of overlying soft tissue.

**Table 1 medicina-58-00640-t001:** Comprehensive literature review of techniques with various key anatomic landmarks used.

Technique	Paper	Key Anatomic Landmark for Docking
Mini-Open Paramedian/Wiltse Approach	Park et al. [8] Marquardt et al. [9] Tessitore et al. [10] Hodges et al. [11] O’Hara and Marshall [12]	Facet Joint and Transverse Processes
Microscopic tubular MIS Approach	Hitchon et al. [13]	Pars Interarticularis
	Phan et al. [14] Siu and Lin [15] Solimon et al. [16]	Facet Joint
	Salame et al. [17] Voyadzis et al. [7]	Cranial Transverse Process

## Data Availability

No new data were created or analyzed in this study. Data sharing is not applicable to this article.

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
