# Peer review of "A Modified Approach for Minimally Invasive Tubular Microdiscectomy for Far Lateral Disc Herniations: Docking at the Caudal Level Transverse Process"

_medicina, 2022, doi:10.3390/medicina58050640_

Round 1

Reviewer 1 Report

Authors reported a good technical note to treat far-lateral disc herniations. Paper was well written. References are correct. Some few points:

  • Figure 3. It not clear in the first surgeon stands on the same part/side of FLDH (left side) or to the other site (right side).
  • In discussion section. About surgical technique, to compare, look at this refs: DOI: 10.1097/BRS.0b013e3181bac710  -  DOI: 10.4103/jcvjs.JCVJS_153_20
  • Could neuronavigation play a role in this type of surgery? discuss.
  • In the conclusion, please state better what this technique has new and its most important advantages.

Author Response

  • Figure 3. It not clear in the first surgeon stands on the same part/side of FLDH (left side) or to the other site (right side).

The figure legend (line 128) was updated to make clear that the surgeon is standing on the right (ipsilateral) side of the disc herniation.

  • In discussion section. About surgical technique, to compare, look at this refs: DOI: 10.1097/BRS.0b013e3181bac710  -  DOI: 10.4103/jcvjs.JCVJS_153_20

Lines 174-177: Contralateral approaches that utilize an inside-out technique requires limited boney removal of the base of the lamina and spinous process, however places the operator at a disadvantage for extraforaminal disc herniations [23]. Extraforaminal lesions, or extreme lateral, should be reached by a far lateral approach described above.

Lines 209-210: Recurrence of an FLDH poses an additional challenge not addressed herein, and optimal treatment is controversial even in conventional revision discectomy vs. fusion [25].

  • Could neuronavigation play a role in this type of surgery? discuss.

Lines 165-170: Docking on the inferior TP also provides readily identifiable anatomy on both x-rays and direct visualization, which decreases the need for more complex and expensive imaging such as intraoperative O-arm computerized tomography navigation [16,22]. If navigation-assistance is utilized, a virtual tip offset of variable length may be used to identify the entry point and surgical trajectory to dock along the caudal TP.

  • In the conclusion, please state better what this technique has new and its most important advantages.

Lines 217-224: Previously, techniques included docking on the facet joint, pars interarticularis, and the cranial TP. Utilizing this reported modification to prior reports requires no further modifications and should allow for easy adoption. The utility of this approach improves upon already established techniques for MIS tubular discectomy for FLDH by minimizing soft tissue dissection, nerve root manipulation, and bone removal, easily identified anatomy on both x-ray and direct visualization. It also eliminates an excess step compared to docking at the cranial TP, which requires subsequent caudal angulation.

Reviewer 2 Report

Thank you for your technical note.  

Can you please comment on modifying the approach (Pars/facet versus inferior TP versus superior TP) based on the relationship of exiting nerve root to the HNP.  Is this relationship important when deciding the docking site?

Author Response

  • Can you please comment on modifying the approach (Pars/facet versus inferior TP versus superior TP) based on the relationship of exiting nerve root to the HNP.  Is this relationship important when deciding the docking site?

Lines 184-188: Additionally, in a cadaveric study it was found that drilling of the caudal TP, or in the case of L5-S1 the superior sacral ala, most easily exposed the exiting nerve root while also provided access to the disc without need for further boney removal [25]. In almost all cases, the extraforaminal disc fragment will be caudal to the exiting nerve root and displace it superiorly and laterally [5].  

Reviewer 3 Report

This technical note is very useful to the spine community.  One question- how does one dock on the transverse process without incurring a fracture of this relatively fragile structure.  It is much different from docking on the lamina or facet, which can withstand more force

Author Response

Reviewer #3: This technical note is very useful to the spine community.  One question- how does one dock on the transverse process without incurring a fracture of this relatively fragile structure.  It is much different from docking on the lamina or facet, which can withstand more force

Lines 83-87: Care must be taken not to use excessive downward force when docking on the TP to avoid incurring a fracture. ‘Walking’ the initial dilator medially and finding the border of the TP-facet junction, similar to a starting point for pedicle screw placement, allows for more downward force and is a useful guide for medial-lateral angulation.